# Development of Sustainable Chemistry in Madagascar: Example of the Valuation of CNSL and the Use of Chromones as an Attractant for Mosquitoes

**DOI:** 10.3390/molecules26247625

**Published:** 2021-12-16

**Authors:** Miarintsoa Michaele Ranarijaona, Ny Aina Harivony Rambala Rakotomena, Mbolatiana Tovo Andrianjafy, Fenia Diane Ramiharimanana, Lydia Clarisse Herinirina, Niry Hasinandrianina Ramarosandratana, Benoit Briou, Pauline Fajardie, Patrick Mavingui, Estelle Métay, Voahangy Vestalys Ramanandraibe, Marc Lemaire

**Affiliations:** 1International Associated Laboratory, Faculty of Sciences, IST Campus, University of Antananarivo-Lyon, Ampasapito, Antananarivo 101, Madagascar; miravo5micha@gmail.com (M.M.R.); nyainaharivony@gmail.com (N.A.H.R.R.); amthta@gmail.com (M.T.A.); miafenia@gmail.com (F.D.R.); irinasoa97@gmail.com (L.C.H.); niryh048@gmail.com (N.H.R.); 2Unité Mixte Processus Infectieux en Milieu Insulaire Tropical, Plateforme Technologique CYROI, University of Réunion, Sainte Clotilde, 97490 La Réunion, France; patrick.mavingui@cnrs.fr; 3Orpia Innovation, Centre National de la Recherche Scientifique CNRS Bâtiment Balard, 34000 Montpellier, France; b.briou@orpiainnovation.com (B.B.); p.fajardie@orpiainovation.com (P.F.); 4Institut de Chimie et Biochimie Moléculaires et Supramoléculaires (ICBMS), University of Claude Bernard Lyon 1/CNRS, 1 rue Victor Grignard, 69100 Villeurbanne, France

**Keywords:** industrial ecology, cashew nut shell liquid, oxyacetic derivative, surfactant, chemical ecology, *Aedes albopictus*, chromone, repellent, attractant

## Abstract

This article describes a part of the results obtained from the cooperation between the University of Lyon1 (France) and the University of Antananarivo (Madagascar). It shows (among others) that useful research can be carried out in developing countries of the tropics if their social, technical, and economic conditions are taken into account. The concepts and methods associated with so-called “green chemistry” are particularly appropriated for this purpose. To illustrate this approach, two examples are shown. The first deals with industrial ecology and concerns waste transformation from the production of cashew nut into an amphiphilic product, oxyacetic derivatives. This product was obtained with a high yield and in a single step reaction. It exhibited an important surfactant property similar to those of the main fossil-based ones but with a much lower ecological impact. The second talks about chemical ecology as an alternative to insecticides and used to control dangerous mosquito populations. New substituted chromones were synthesized and showed biological activities toward *Aedes albopictus* mosquito species. Strong repellent properties were recorded for some alkoxylated products if others had a significant attractant effect (Kairomone) depending on their stereochemistry and the length of the alkyl chain.

## 1. Introduction

Most developing countries are located in the tropics. Generally, chemical industrial technologies are not adapted to the climatic, social, and economic conditions of these areas. High temperatures, humidity, or conversely, lack of water, and the relatively poor supply of electricity, solvents, and chemicals often drive the development of classical chemistry in developing countries. Indeed, “modern” chemistry using toxic and dangerous solvents and reagents, at high or very low temperature, is only possible in developed countries where risks can be controlled and where pollution control technologies are available. The principles of Green chemistry avoid dangerous chemicals (3° less hazardous chemical synthesis, 4° designing safer chemicals, 5° safer solvents and auxiliaries) and process (12° inherent safer chemistry for accident prevention). Those are exactly what is required if chemical transformations have to be performed in tropical areas. Furthermore, the availability of an abundant biomass with a very high biodiversity in tropical areas (7° use of renewable raw materials and 10° design for degradation) should facilitate the development of sustainable chemistry. Finally, the absence of a former chemical industry and the high demand for specialty chemicals of a fast-growing population promote the creation of new, and harmless chemical company in these regions. Almost ten years ago, these observations led the University of Lyon1 (France) and University of Antananarivo (Madagascar) to create a joint laboratory in Antananarivo to perform research using appropriate technologies and raw materials available in Madagascar. We are developing three research topics: 1° the valorization of agro-industry wastes; 2° the use of chemical ecology as an alternative to insecticides; and 3° the search for herbal medicine to treat emerging infectious diseases. The first two are achieved mainly by adapted and existing technologies in Madagascar and are the subject of this article. All these works were carried out mainly in Madagascar with the help of the University of Lyon1 and the University of La Reunion. In this article, we describe new results on the valorization of cashew nut shell liquid and on chemical ecology research to control the population of *Aedes albopictus* mosquitoes responsible for the transmission of chikungunya and dengue in Indian Ocean Islands [1,2].

Cashew trees are used for reforestation in many African countries and the production of cashew almond is steadily growing thanks to local and developed countries’ demand. Use of non-renewable starting materials is one of the main problems of modern industry [3]. The research for alternatives to these products is flourishing, especially by using agriculture wastes.

Cashew nut shell liquid (CNSL) is a byproduct of the cashew industry [4] obtained by extracting cashew nut shells [5]. This raw material is potentially available at the scale of hundred thousand tons. The crude CNSL consists mainly of anacardic acid (**1**) (65%) cardol (**2**) (20%), and cardanol (**3**) (10%), [6,7]. Several types of surfactants have already been synthesized from its constituents: polyethoxylates, sulfonates, and carboxylates [8,9,10,11]. In most of the cases, these products are obtained through a multistep synthesis and after tedious and expansive separation of the mixture.

In this study, carboxylate surfactants were synthesized from separate components (anacardic acid (**1**) and cardol (**2**)) and in parallel from crude CNSL in a one-step synthesis. To reach this goal, the reaction parameters were optimized (basic catalyst, reagents, solvent, stoichiometry, and reaction time) and physico-chemical and applications tests such as water solubility, Critical Micellar Concentration (CMC)values, Hydrophilic Lipophilic Balance (HLB) value, toxicology and ecotoxicology studies, foaming power, and detergent powers were performed.

In the field of chemical ecology, the selective control of insect populations is one of the main problems of sustainable development. Limiting the population of certain insect pests is necessary for the protection of humans, livestock, and crops, but many insects also play an important role in ecological balance. On the one hand, more effective and more selective insecticides are discovered and available on the market; on the other hand, the side effects are always present and induce application difficulties [12,13]. In addition, the rapid appearance of resistance from the target insects makes the strategy globally unsustainable [14]. Finally, consumers are also asking for new approaches that respect both the environment and economic development. From this point of view, chemical ecology could be one of the main tools to achieve this important goal. Research into new attractants is generally limited to natural products existing in nature, either on the skin of the prey (isovaleric acid, 3 octenol, etc.) or in plant emissions. We believe that finding new attractants, with high selectivity for dangerous species, could be an important tool for creating selective traps to limit populations of dangerous species without destroying useful ones. We have already shown that hydroxycoumarins are selective attractants for *Aedes albopictus* and *Anopheles* vectors of malaria [15,16]. We are currently working on a new family of semiochemicals: substituted chromones which have potential attractive and repellent properties. Chromones are described to have a various biological activities; their synthetic accessibility and structural diversity make them excellent templates for structural modifications [17,18,19].

## 2. Materials and Methods

### 2.1. Cashew Nut Shell Liquid Valorization as Surfactant

#### 2.1.1. Materials

Cashew nut shells were collected from operators in the north of the island (Antsiranana) and were used as a source of CNSL. Sodium chloroacetate, ethyl chloroacetate, potassium hydroxide, activated carbon (FE12510F), and pyrene were purchased from Sigma Aldrich, WWR, and CARLO ERBA. Sodium hydroxide, sodium bicarbonate, and solvents (hexane, ethyl acetate, EtOH, MeOH) came from a local chemical company (Chimidis). All reagents were used without purification.

#### 2.1.2. Extraction of CNSL

Into a Soxlhet, 500 g of crushed shells were introduced for percolation extraction. Then, 1 L of hexane was poured in a 2 L round bottom three-necked flask and 500 mL in a Soxlhet device, at the top of which a refrigerant is attached. A heating system was used to bring the hexane to a boil. The extraction lasted 10 h and after the last syphon, the extract was recovered, filtered, and evaporated.

#### 2.1.3. Separation of CNSL Constituents

CNSL constituents could be separated by distillation but this process promotes the transformation of anacardic acid (**1**) into cardanol (**3**) which would be the main component [20]. However, this molecule is monofunctionalized which drastically limits its applications. For our study, anacardic acid (**1**) was separated from the other constituents of CNSL by treatment with calcium hydroxide [21]. In a mixture of EtOH/H_2_O (4/2, *v*/*v*), 20 g of CNSL and 10 g of calcium hydroxide were dissolved. They were stirred at 50 °C for 24 h. The resulting salt (calcium anacardate) was filtered, washed with ethanol, and dried. It was then treated with 6N HCl until its complete dissolution. The reaction medium became acidic and liquid–liquid separation with ethyl acetate was carried out to recover the desired product: 14 g (70%) of anacardic acid (**1**). The ethanol solution was neutralized, evaporated, and was separated on a column chromatography to give a brown viscous oil [22] containing 3.2 g (16%) of cardol (**2**) and 0.8 g (4%) of cardanol (**3**) (Figure 1).

#### 2.1.4. Oxyacetic Acid of Anacardic Acid (**5**) Synthesis

Anacardic acid (**1**) (10 mmoL, 3.42 g), NaOH (20 mmoL, 0.8 g) and 10 mL EtOH/H_2_O (1/1, *v*/*v*) were mixed in a 50 mL flask. The temperature was increased to 70 °C to dissolve the NaOH. Then, 1.8 mmoL of sodium chloroacetate was added. The reaction was left for 24 h. After cooling, 3 mL of 2N HCl solution was introduced until a pH equal to 3. The diacid was recovered in 20 mL of ethyl acetate and 2 g of activated carbon (FE12510F) was added. The mixture was stirred for 5 h at room temperature. After filtration and washing (3 times) of the activated carbon with ethyl acetate, the filtrate was evaporated to yield 79% of oxyacetic acid of anacardic acid (**5**). It is a light brown pasty product.

Oxyacetic acid of anacardic acid (**5**): proton NMR (300 MHz, CDCl_3_, TMS): δ (parts per million (ppm)) 7.25–6.7 (m, 3H); 5.6–5 (m, 5H); 4.7 (s, 2H); 2.6 (m, 2H); 2.4–1.1 (m, 12H); 0.9 (m, 3H).

#### 2.1.5. Dioxyacetic Acid of Cardol (**6**) Synthesis

Cardol (**2**) (10 mmoL, 3.18 g), KOH (20 mmoL, 1.18 g), and 10 mL of EtOH were mixed in a 50 mL flask. The flask was raised to 70 °C; after NaOH solubilization, 24 mmoL of sodium chloroacetate was added. The reaction was left for 24 h. After cooling, 5 mL of 2N HCl solution was introduced until a pH equal to 3. The product was recovered in 20 mL of ethyl acetate and 2 g of activated carbon (FE12510F) was added. The mixture was stirred for 5 h at room temperature. Then, filtration and washing (3 times) of the activated carbon with ethyl acetate were realized. Finally, the filtrate was evaporated to give the dioxyacetic acid of cardol (**6**). It is a brown waxy product obtained with 82% yield.

Dioxyacetic acid of cardol (**6**): proton NMR (300 MHz, CDCl_3_, TMS): δ (parts per million (ppm) 6.45–6.29 (m, 3H); 5.7–5 (m, 5H); 4.66 (s, 2H); 2.8 (m, 2H,); 1·3–2.6 (m, 12H,); 0.9 (m, 3H).

#### 2.1.6. Oxyacetic Acids of CNSL (**7**) Synthesis

Taking into account that anacardic acid (**1**) is the major constituent of the CNSL, the calculation was made in relation to it. CNSL (3.42 g, approximately 10 mmoL), KOH (20 mmoL, 1.18 g), and 10 mL of EtOH were mixed in a 50 mL flask. The flask was closed and raised to 70 °C; after total solubility of KOH, 18 mmoL of sodium chloroacetate was added. The reaction was left for 24 h. After cooling, an acidification with 4.3 mL of 2N HCl solution was performed to obtain pH equal to 3. The diacid was extracted in 20 mL of ethyl acetate and 2 g of activated carbon (FE12510F) was added. The mixture was stirred for 5 h at room temperature. Then, a filtration and washing (3 times) of the activated carbon with ethyl acetate were realized. Finally, the filtrate was evaporated to give the oxyacetic acids of CNSL (**7**). It is a light brown pasty product obtained with a yield 76% after discoloration treatment.

Oxyacetic acid of CNSL (**7**): proton NMR (300 MHz, MeOD, TMS): δ (parts per million (ppm) 6.7–7.4 (m, 3H); 6.5–6.3 (m, 3H); 5.8–4.8 (m, 15H); 4.6–4.8 (s, 6H, 2.7) (m, 6H); 1.3–2.4 (m, 36H); 0.9 (m, 9H).

Oxyacetic acid of CNSL (**7**): carbon NMR (300 MHz, MeOD, TMS): δ (parts per million (ppm) 170 (6C); 155 (4C); 140 (3C); 137, 122, and 110 (10C); 130–126 (12C), the positive peak at 114 (6C); 65–60 (4C); 36–25 (27C); 12.6 (3C).

Oxyacetic acid of CNSL (**7**): carbon DEPT 135 NMR (300 MHz, MeOD, TMS): δ (parts per million (ppm) 137, 122, and 110 (10C); 130–126 (12C), the positive peak at 114 (6C); 65–60 (4C); 36–25 (27C); 12.6 (3C).

#### 2.1.7. Physicochemical, Biological, and Application Tests

The tests solubility of CNSL derivatives in water and in hexane was carried out by mixing 100 mg of products in 1 mL of water and then by evaporating the soluble part to deduce soluble products weight. The experiment was repeated three times to get a significant average.

To obtain the carboxylate surfactant, which is sodium oxyacetate of CNSL (**8**), a salt formation with baking soda was carried out.

The CMC value of this surfactant was determined with a Wilhelmy plate and by fluorescence emission of pyrene.

Ecotoxicity test was performed with 10 nauplii 24 h-old (*Artemia catvis* cysts). In a hemolysis tube containing 1 mL of artificial seawater and nauplii, 0.1 mL of stock solutions such as CNSL derivative, LABSA (linear sodium alkyl benzene sulfonate), and SDS (sodium dodecyl sulfate) at pH 8 were injected every 15 min.

Skin toxicity of CNSL derivative, LABSA, and SDS were investigated for the viability of normal human dermal fibroblasts (NHDF) and normal human epidermal keratinocytes (NHEK) using a standard WST-8 reduction test at Bioalternatives in France.

Foaming power was assessed and measured according to the standard (NFT 73–404) [23] by comparing with commercial surfactants such as LABSA (and SDS). Their pHs were equalized to 8. The three surfactant solutions (10 g/L) were mechanically shaken for 10 min and the foam height was measured. After that, they were used to wash oil-soiled wipes to compare their detergent power at room temperature with identical mechanical stirring.

#### 2.1.8. Method of Analysis

During this study, thin layer chromatography (TLC) was used as the main technology to determine the conversion and the purity of products. This technology was one of the main tools until the 1970s in developed countries. Now, it is generally replaced by LC coupled with mass spectroscopy or NMR, which are currently available in most the academic laboratories. It is not the case in Madagascar but TLC is still a very powerful technology allowing qualitative and semi-quantitative evaluation when “authentic samples” are available. Reactions were monitored on TLC using 60 F254 silica gel plates on a MERCK aluminum substrate and by HPLC consisting of a Shimadzu LC-10 AS spectrometer, an SPD-6A Shimadzu UV detector, and a C-R 6A Shimadzu recorder. The TLC was made with an eluent system, ethyl acetate 100%, and revealed with vanillin sulfuric. HPLC was carried out according to the following conditions: Column Rp-18 (4.6 mm 250 mm Grace Brand, made in USA), eluent system: MeOH/Ethyl acetate (7/3), Rev: UV 280 nm, C = 5 mg/mL. NMR and mass spectroscopy analysis were carried out at ICBMS Lyon1 University. The NMR was done with a single-dimensional NMR spectrometer ^1^H and ^13^C BRUKER 300 MHz in the MeOD and CDCl_3_.

### 2.2. Chemical Ecology as Alternative to Insecticides

#### 2.2.1. Synthesis of Substituted Chromones

All chemicals used are commercial products supplied by Sigma Aldrich. The 7-hydroxychromone was used as a starting product for all the reactions. Thin layer chromatography was performed to follow the evolution of each reaction and to analyze the purity of each product. Column chromatography was carried out on silica gel 60 (230–400 mesh ASTM) with hexane/ethyl acetate as solvent at various proportions specified in the corresponding experiment. A Bruker spectrometer (300 MHz both for ^1^H and ^13^C), deuterated chloroform (CDCl_3_) was used at ICBMS Lyon1 University for the recording of proton (^1^H) and carbon (^13^C) NMR spectra. A Bellingham and Stanley ADP 220 polarimeter, at 25 °C with a 0.2% dilution in EtOH, was employed in our laboratory for the measurement of the optical rotation.

Two methods were used to synthesize all the chromone derivatives: the first one was phase transfer catalysis and the second the Mitsunobu reaction.

##### General Procedure for the Synthesis of Chromones Derivatives by Using Phase Transfer Catalysis Reaction

The phase transfer catalysis is a known technique which can be used for reaction of *O*-alkylation employing carbonates as bases and several phase transfer catalysts such as tetrabutylammonium bromide (TBAB), tetrabutyl ammonium hydrogen sulfate (TBAHS) [24]. For our study, the following general method was used for the synthesis of chromones derivatives. 7-hydroxychromone (**9**) and bromoalkane were added in a mixture of toluene and K_2_CO_3_ (5 equiv.), then TBAHS (5–50% by mass of the starting material) were added. The reaction was brought to reflux at 105 °C from 2 to 6 h. After, the mixture was filtrated and evaporated. Purification on silica gel chromatography using hexane/ethyl acetate at different proportions as solvent, afforded the expected compound (Figure 1).

Likewise, the synthesis of 7-*sec*-butoxychromone (**10**), 7-*sec*-pentoxychromone (**11**), 7-*sec*-nonyloxychromone (**12**), and 7-(2′-ethyl)hexyloxychromone (**13**) were carried out using the same method as shown in Table 1.

##### General Procedure for the Synthesis of Chromone Derivatives by using Mitsunobu Reaction

The Mitsunobu reaction is used to transform an alcohol to a variety of functional groups such as ether, using triphenylphosphine and an azodicarboxylate such as diethyl azodicarboxylate (DEAD). This reaction makes it possible to obtain a pure ether enantiomer from a pure alcohol enantiomer with configuration inversion (Figure 2) [25,26].

In a 50 mL reactor equipped with a thermometer, a magnetic stirrer, and a slight nitrogen overpressure, dichloromethane, 7-hydroxychromone (**9**) were added successively, then 2 equiv. of chiral alcohol and 2 equiv. of triphenylphosphine. The mixture was stirred a few minutes, and then 0.8 equiv. of diethylazodicarboxylate (DEAD) was introduced dropwise at room temperature. The reaction mixture was stirred at room temperature for 15–24 h, then filtrated and evaporated. The mixture was purified by column chromatography with silica gel using hexane/ethyl acetate (8/2, *v*/*v*) as eluent. Enantiomers (*R*) and (*S*) of 7-*sec*-butoxychromone and 7-*sec*-pentoxychromone were synthesized by the Mitsunobu reaction. The conditions of synthesis are shown in Table 2.

#### 2.2.2. Chemical Analysis

^1^H and ^13^C NMR analyzes were carried out using BRUKER spectrometer (Billerica, MA, USA). The optical rotation was measured at 25 °C on an ADP 220 polarimeter (Bellingham, England) with a 0.2% dilution in EtOH. Three replicates were done for each product (Appendix A, Appendix A).

#### 2.2.3. Mosquitoes

Larvae and pupae of *Aedes albopictus* were collected around the city of Antananarivo, Madagascar. The specimens were reared in the insectary of the International Associated Laboratory located at the Ampasapito campus, Antananarivo, under the following conditions: ±25 °C temperature, ±70% relative humidity, and 12/12 h photoperiod. The larvae were placed in tanks filled with breeding water. Pupae were separated from the larvae and transferred to jars containing a small amount of breeding water. The jars containing the pupae were put inside gauze cages for the emergence of adults. Dog biscuit powders rich in tetramin^®^ were given to the larvae as food while sucrose solution 6% meals were served to male and female adults, when needed females were allowed to take blood meals for gonotrophic development. Females adults from F2 generation were used for the bioassay.

#### 2.2.4. Bioassay

The evaluation of the repellent and attractive activities of the products towards mosquitoes was performed in the laboratory on the tunnel olfactometer (glass tube 7 × 7 cm side and 1.20 m long) [27]. This device is divided into three zones: a neutral zone which is located in the middle, a treated zone which is situated on the product deposition side, and a control zone located on the blank deposit side. The tunnel olfactometer has three openings: two on each side allowing the introduction of the paper impregnated with the product to be tested and on the other the paper containing a control. The opening in the middle allows the introduction of mosquitoes. These openings are closed with lids during the test. A total of 15 female mosquitoes (nulliparous and aged 5 to 12 days) were introduced into the neutral zone of the olfactometer. They were left there for 10 min for an adaptation time. The impregnated papers (treated and control) were introduced, and the barriers were opened to allow mosquitoes to move freely inside the device. Three replicates were performed for each product. The results were recorded every 5 min and the duration of test was 20 min.

#### 2.2.5. Analysis of Data

The values recorded were the activity index (AI) (Equation (1)) and the repellency index (RI) (Equation (2)). The (AI) describes the percentage of mosquitoes in the control zone (T) and the treated zone (P) compared to the total number of mosquitoes tested. This value must be greater than 30% for the test to be considered significant. Repellency index (RI) represents the percentage of the difference in the number of mosquitoes in the control zone and the treated zone divided by the sum of these two values. A negative value of the parameter (RI) indicates an attractant activity (Kairomone index, (KI)) of the product tested (Equation (3)).
(1)$$\mathrm{AI}=\left(\frac{T+P}{15}\right)\times 100$$
(2)$$\mathrm{RI}=\left(\frac{T-P}{T+P}\right)\times 100\;$$
(3)$$\mathrm{KI}=\left(\frac{P-T}{T+P}\right)\times 100$$

Graphs and data analysis were performed using Graphpad prism software. *t*-test analysis was carried out to compare the mean number of mosquitoes in the treated and control zones. Analysis of variance by ANOVA was done to compare the repellency or kairomone indices of the different doses or quantities of each product. Standard deviation (SD) was used in tables and graphs to observe the reliability and reproducibility of the results and the confidence interval was estimated at 95%.

#### 2.2.6. Products Tested

A 5% (50 mg/mL) ethanolic solution of each product was prepared. Then, 100 µL of this solution were impregnated on the filter paper for a 5 mg deposit. Two 100 µL deposits correspond to the 10 mg quantity and so on for the other quantities used. For the combination of two products, 25 mg of each product were mixed in 1 mL of ethanol to have a 50 mg/mL solution. In total, nine synthesized compounds were tested and four known products were tested as positive reference, including two repellents and two attractants. The tested quantities of which are presented in Table 3.

## 3. Results and Discussions

### 3.1. Cashew Nut Shell Liquid Valorization as Surfactant

#### 3.1.1. CNSL Extraction

CNSL is an oily black brown product. Extraction with hexane gave 37% CNSL. It gives 70% of anacardic acid (**1**), 16% of cardol (**2**) and 4% of cardanol (**3**) isolated after precipitation with calcium hydroxide and column chromatography. This is very close to what is described by O. Victor-Oji et al. [28]. In some cases, cashew nut shells were collected over 3 years ago and the results turned out to be very similar, which shows the very high stability of this raw material.

#### 3.1.2. Oxyacetic Acid of Anacardic Acid (**5**) and Optimization

In order to obtain optimal conditions for crude CNSL transformation, we firstly optimized the reaction with pure anacardic acid (**1**).

Synthesis of oxyacetic acid of anacardic acid (**5**) was already described by Sonia Koteich Khatib et al. It consists of etherifying the anacardic acid (**1**) at the hydroxyl group level in the presence of KOH in a solvent mixture (toluene and DMSO) and then treated by 10% of sulfuric acid solution to obtain the oxyacetic acid of anacardic acid (**5**) crude in 92% yield, pH = 5. After neutralization, a salt disodium oxyacetate of anacardic acid was obtained. According to these authors, the products formed have surfactant and antibacterial properties [29]. This approach is particularly effective, but has a drawback: solvents (toluene and DMSO) are relatively expensive and toxic.

#### 3.1.3. Oxyacetic Acid of Anacardic Acid (**5**) Synthesis with Ethyl Chloroacetate

In our study, the reaction was carried out with ethyl chloroacetate in the presence of potassium carbonate and a small amount of quaternary ammonium (tetrabutylammonium acid sulphate) as a phase transfer catalyst. Potassium carbonate is used to deprotonate the hydroxyl functions of the starting product [30] while quaternary ammonium is used to perform solid/liquid phase transfer [31]. Ethyl chloroacetate is used both as a reagent and as a solvent, so we put it in excess of 10 mol/mol. A good yield (>90%) of ester oxyacetic of anacardic acid (**4**) was obtained without purification and excess ethyl chloroacetate was recovered after distillation (recovering yield 70%). Oxyacetic acid was then obtained after hydrolysis in the presence of sodium hydroxide with a quantity of 2.2 mol/mol compared to anacardic acid (**1**) in an EtOH/H_2_O (1/1 *v*/*v*) mixture solution, at 80 °C for 12 h. After cooling, the reaction medium was acidified by 2N HCl until pH = 3. The oxyacetic acid of anacardic acid (**5**) was recovered in the ethyl acetate after liquid–liquid extraction with 75% yield. This method requires a two-step transformation (etherification and hydrolysis). In addition, it induces the formation of potassium salt and the loss of an ethanol molecule (Figure 3).

#### 3.1.4. Oxyacetic Acid of Anacardic Acid (**5**) Synthesis with Sodium Choroacetate

This time we used solvent with low toxicity. Since NaOH and sodium chloroacetate are only soluble in water, we used an aqueous solvent mixture (EtOH/H_2_O). In this solvent, with 2 mol/mol of NaOH instead of potassium hydroxide and 1.2 mol/mol of sodium chloroacetate relative to anacardic acid (**1**), complete conversion was never obtained even after 72 h of reaction. After cooling, acidification with 2N HCl was carried out to reach a pH = 3. Oxyacetic acid of anacardic acid (**5**) was recovered by liquid–liquid extraction with ethyl acetate/H_2_O with a crude yield of 93% and a proportion of desired product of 70%, after solvent distillation. This synthetic route is interesting insofar as we do not use costly and toxic solvent. However, the longer reaction time (72 h) is a major drawback (Figure 4).

Optimization is required to reduce the duration of the reaction. To do this, some parameters have been changed such as the nature of the base, solvent, and stoichiometry.

#### 3.1.5. Optimization

##### Nature of the Base

We compared the effects of sodium hydroxide (in pellets, then in solution), potassium carbonate (in powder), and potassium hydroxide in pellets (Table 4). The results showed that potassium hydroxide was the best, mainly due to its greater solubility in ethanol.

##### Solvent and Temperature

Since sodium chloroacetate is only soluble in polar solvents, we used water, EtOH/H_2_O (1/1, *v*/*v*), pure ethanol, and acetone. It was found that the reaction medium is homogeneous using water, but nucleophilic substitution was too slow. With the EtOH/H_2_O (mixture 1/1, *v*/*v*), the reaction took place but it was also too slow. With ethanol, it was acceptable (conversion almost complete >90% after 24 h). With acetone, the reaction medium was never homogeneous even in the presence of a little quaternary ammonium and the conversion very low (Table 5).

Temperature was chosen in order to avoid ebullition and loose of solvent, therefore 70 °C was chosen considering that Antananarivo is at an altitude of 1300 m.

##### Stoichiometry and Reaction Time

For stoichiometry, we only varied that of sodium chloroacetate and not that of the base to avoid the solvolysis of chloracetate. Calculation of stoichiometry was done using anacardic acid (**1**) molecular weight even when using natural mixture, and we varied the ratio of sodium chloroacetate/anacardic acid (**1**) from 2.4 mol/mol to 1.3 mol/mol (Table 6).

When using 1.8 mol/mol of sodium chloracetate per mol of anacardic acid (**1**) we could reduce the reaction time to 24 h without decreasing the yield.

The best stoichiometry is 1.8 mol/mol sodium chloroacetate relative of anacardic acid (**1**) in ethanol potassium hydroxide solution at 2 mol/L. A conversion > 90% was observed and an isolated yield of 96% (Table 7).

#### 3.1.6. Oxyacetic Acid of Cardol (**6**)

As the reaction with anacardic acid (**1**) was optimized, the parameters found in the optimization were reproduced with cardol (**2**) taking into account the number of active sites. A yield of 82% was found after treatments (Figure 5).

#### 3.1.7. Oxyacetic Acid of CNSL (**7**) and Sodium Oxyacetates of CNSL (**8**)

Since our objective was to value the raw mixture of CNSL and to avoid the separation step and purification. We performed the synthesis reaction of the oxyacetic derivatives from the mixture using sodium chloroacetate. The optimal conditions outlined above were applied with CNSL (**7**) taking into account only the main product: anacardic acid (**1**). Potential traces of sodium chloroacetate were eliminated by modifying the pH at the end of the reaction with a solution of HCl (pH 3–4), then the product was extracted with water and ethyl acetate. Chloroacetic acid is highly soluble in water. Oxyacetic acid of CNSL (**7**) was obtained as a dark brown product in the organic phase. A discoloration of the diacid with 50% of carbon activated (FE12510F) in the ethyl acetate was then carried out. They were mixed and stirred at room temperature for 5 h. After filtration, several washes of the activated carbon with ethyl acetate, and evaporation of the filtrate, the product was obtained in the form of a light brown wax with a yield of 76% (Figure 6). The activated carbon was washed with several solvents (water, MeOH, ethyl acetate, and hexane) and reused without loss of activity.

#### 3.1.8. Physicochemical, Biological, and Application Tests

##### Solubility of Oxyacetic Acid Derivatives

Solubility of oxyacetic acid of CNSL (**7**) is higher than pure oxyacetic acid of anacardic acid (**5**) in both in water and hexane. Generally, mixtures of isomers or products of similar structures exhibit a much lower melting point and higher solubility than pure isolated products (Table 8).

Oxyacetic acid of CNSL (**7**) is soluble in both water and hexane; however, these values are low for a valorization in the field of surfactant. Salt formation with baking soda leads to completely soluble in water and very foaming products.

##### CMC and HLB Value of Sodium Oxyacetate of CNSL (**8**) Surfactants

Surface tension was measured three times for each solution using the method with a Wilhelmy plate. The average is calculated. Afterwards, the graph surface tension versus concentration is plotted. CMC corresponds to the intersection of the two trend lines. The surface tension of sodium oxyacetate of CNSL (**8**) is approximately 35 mN/m. This value is similar to surface tension of commercial surfactants (30 to 40 mN/m for LABSA and SLES). CMC of sodium oxyacetate of CNSL (**8**) appears to be 0.13 mM. This value is low compared with those of LABSA, SLES, and SDS, which are, respectively, in the order of 1.33 Mm, 1.01 mM, and 2.7 mM [32,33]. This value was verified on the fluorescence emission spectrum of pyrene. This was recorded 3 times for each solution. The energy value for two wavelengths (I1 = 373 nm and I3 = 384 nm) were acquired and the ratio I1/I3 was calculated. Eventually, the graphs I1/I3 = f(log(C)) were plotted (Figure 2). CMC value corresponds to the intersection of the two trend lines.

The results for sodium oxyacetate of CNSL (**8**) surfactant is close to 0.1 mM similar to that obtained using the previous method. Indeed, the sodium salts of the CNSL oxyacetic derivative have strong surfactant effect at low concentration compare to LABSA, SLES, and SDS. This may be ascribed to the long hydrophobic chain and of the large polar head. Their HLB values derived from the formula proposed by Davies confirmed this approach [34] (Equation (4)).
(4)HLB=7+∑hydrophilic groups increment−∑hydrophobic groups increment

The HLB increments of the different groupings are shown in the following Table 9.

Results were 36 for sodium oxyacetate of CNSL (**8**) (calculated on the structure of the majority product), 30 for LABSA, and 33 for SDS.

##### Ecotoxicology Study

It is necessary to evaluate the acute and chronic toxicity of this new surfactant on aquatic animals due to the production of waste-water when they are used for detergency [35]. Aquatic toxicity was tested with nauplii (*Artemia catvis* cysts). Several concentration ranges for each stock solution (sodium oxyacetate of CNSL (**8**), LABSA, and SDS) were prepared such as 0.25 g/L, 0.5 g/L, 1 g/L, 2.5 g/L, 5 g/L, and 10 g/L. The test was performed by injecting 0.1 mL of stock solutions every 15 min into a hemolysis tube containing 10 24 h-old nauplii and 1 mL of artificial seawater. Table 10, Table 11 and Table 12 show the results.

LABSA is relatively toxic at pH 8, its lethal concentration 50 is 0.17 g/L and LC_90_ is equal to 0.22 g/L. The SDS is moderately aquatoxic with a lethal concentration 50 of 0.71 g/L and LC_90_ to 1.11 g/L. Sodium oxyacetate of CNSL (**8**) is by far the least aquatoxic of the three surfactants tested with higher lethal concentrations of 3.33 g/L (LC_50_) and 4.12 g/L (LC_90_).

##### Cytotoxicity on Viability of Dermal Fibroblasts and Normal Human Epidermal Keratinocytes

Three products were tested: sodium oxyacetate of CNSL (**8**), LABSA or linear alkyl benzene sodium sulfonate, and SDS or sodium dodecyl sulfate. Sodium oxyacetate of CNSL (**8**) at the highest concentration of 1 mg/mL did not affect the viability of NHDF and NHEK.

##### Effect on Viability of NHDF (Normal Human Dermal Fibroblasts)

A slight reduction in viability (85% of the control), but without morphological changes in cells, was observed at 1 mg/mL. However, from the next dilution (0.5 mg/mL) to the lowest concentration (4.57 × 10^−1^ µg/mL), the viability of NHDF was not affected by CNSL oxyacetate.

LABSA was solubilized the same way in DMSO and very strong reduction in viability (cytotoxicity of the compound) was observed up to 0.206 mg/mL. The lower concentrations, 0.069 and 0.023 mg/mL, showed no cytotoxic effect.

SDS, soluble at 100 mg/mL in the test medium showed a strong reduction in viability of NHDF up to 0.123 mg/mL. The first non-toxic concentration with no morphological changes was 0.041 mg/mL.

##### Effect on Viability of NHEK (Normal Human Epidermal Keratinocytes)

Compared to NHDF, NHEK cells are more fragile in culture and generally more sensitive to the compounds tested, so the first concentration on NHEK was selected based on the viability results obtained on NHDF.

As LABSA had a strong impact on the viability of NHDF, it was tested from 7.6 µg/mL on NHEK. At this concentration and up to 3.14 × 10^−2^ µg/mL, it induced high toxicity and/or morphological changes on NHEK. Tested at lower concentrations (1.05 × 10^−2^ µg/mL and 3.48 × 10^−3^ µg/mL), the LABSA no longer affected cell viability.

Similarly, based on the results obtained with NHDF, the SDS effect on NHEK viability was tested from 1.1 mg/mL. The toxicity observed was significant. The first concentration that did not induce any reduction in viability or morphological changes was 1.52 µg/mL.

CNSL sodium oxyacetate has little or no toxicity to skin cells, much less than the most common industrial surfactants, and its use in the field of surfactants presents no risk (Table 13).

##### Foaming Property and Cleaning Powers

Foaming property has been tested according to standard (NFT 73–404). This is a static method by Ross–Miles [28]. It involves evaluating the volume of the initial foam and its stability over time. Sodium oxyacetate of CNSL (**8**) validated this test. Thus, in order to verify its competition with petro-sourced surfactants, a comparison of its foaming power (sodium oxyacetate of CNSL (**8**)) with LABSA (linear alkyl benzene sodium sulfonate) and SDS (sodium dodecyl sulfate) was carried out and the results are given by the following curve (Figure 3).

Sodium oxyacetate of CNSL (**8**) is foaming similarly to LABSA and SDS, their detergent power has also been studied.

A detergent must have a maximum detergent efficiency and be adapted to the soiling to be eliminated [36]. Comparison of their detergent power with that of LABSA and SDS was carried out. To do this, three tissues contaminated with drain oil were prepared 24 h before the test. Then, we took the three solutions (sodium oxyacetate of CNSL (**8**), LABSA, and SDS) at 10 mg/mL and pH 8 to wash the wipes at room temperature and with stirring for 30 min. The results before and after each wipe are shown in Figure 4.

The cleaning powers of the three solutions are similar, with a slight advantage of our product from CNSL (sodium oxyacetates of CNSL (**8**)). Thus, our surfactant is also a good detergent.

### 3.2. Chemical Ecology

#### 3.2.1. Synthesis of Substituted Chromones

In total, four products were synthesized by phase transfer catalysis reaction: 7-*sec*-butoxychromone (**10**), 7-*sec*-pentoxychromone (**11**), 7-*sec*-nonyloxychromone (**12**), and 7-(2′-ethyl)hexyloxychromone (**13**). The yield and conversion of each reaction were different. The results of each reaction are shown in Table 14.

Yields were moderate to good for butoxy, nonyl, and 2′-ethyl-hexyl chromone; lower yield was obtained with pentoxychromone due to purification difficulties.

Four enantiomers were synthesized by the Mitsunobu reaction: *R*-(−)-7-*sec*-butoxychromone (**14**), *S*-(+)-7-*sec*-butoxychromone (**15**), *R*-(−)-7-*sec*-pentoxychromone (**16**), and *S*-(+)-7-*sec*-pentoxychromone (**17**). The results are presented in Table 15.

In all cases, we obtained moderate yields, indicating the power of the Mitsunobu reaction, which nevertheless has a very poor atom economy.

For phase transfer catalysis, a large excess of alkyl bromide was required (6 equiv. of bromoalkane) and gave a good yield as seen in the synthesis of the 7-*sec*-butoxychromone (**10**) (R = 73%). When the quantity of reagent was reduced (3 equiv. of bromoalkane), the yield of the reaction became lower (R = 33%). A high concentration of catalyst gave a better yield. In the synthesis of 7-*sec*-nonyloxychromone (**12**), using 50% by mass of tetrabutyl ammonium hydrogen sulfate afforded good yield (R = 92%) and a shorter reaction time. The two technologies, i.e., phase transfer catalysis and Mitsunobu reaction were proven to be usable in our laboratory in a tropical country although they do not afford good atom economy; their main advantages are the reagents which are easy to handle and have a relatively low toxicity.

#### 3.2.2. Chemical analysis of products

*7-sec-butoxychromone* (**10**), NMR ^1^H (300 MHz, in CDCl_3_) δ 8.11 (1H, d), 7.77 (1H, d), 6.96 (1H, dd), 6.82 (1H, s), 6.26 (1H, d), 4.41 (1H, m), 1.77 (2H, m), 1.36 (3H, d), 1 (3H, t); NMR ^13^C (300 MHz, in CDCl_3_) δ 177.05, 162.90, 158.31, 154.83, 127.18, 118.40, 115.49, 112.83, 101.82, 75.82, 28.99, 18.99, 9.68.

*7-sec-pentoxychromone* (**11**), NMR ^1^H (300 MHz, in CDCl_3_) δ 8.13 (1H, d), 7.75 (1H, d), 6.9 (1H, dd), 6.8 (1H, s), 6.28 (1H, d), 4.5 (1H, m), 1.75 (2H, m), 1.6 (3H, d), 1.38 (2H, m), 0.90 (3H, t); NMR ^13^C (300 MHz, in CDCl_3_) δ 177.03, 162.90, 158.31, 154.79, 127.19, 118.40, 115.45, 112.83, 101.82, 74.42, 38.31, 19.44, 18.64, 13.95.

*7-sec-nonyloxychromone* (**12**), NMR ^1^H (300 MHz, in CDCl_3_) δ 8.09 (1H, d), 7.75 (1H, d), 6.93 (1H, dd), 6.8 (1H, s), 6.28 (1H, d), 4.46 (1H, m), 1.76 (2H, m), 1.62 (3H, d), 1.41 (2H, m), 1.34 (2H, t), 1.34 (2H, t), 1.28 (2H, t), 1.28 (2H, t), 0.87 (3H, t); NMR ^13^C (300 MHz, in CDCl_3_) δ 177.01, 162.88, 158.30, 154.77, 127.18, 118.40, 115.44, 112.84, 101.77, 74.69, 36.18, 31.75, 29.46, 29.18, 25.40, 22.61, 14.05.

*7-(2′-ethyl)hexyloxychromone* (**13**), NMR ^1^H (300 MHz, in CDCl_3_) δ 8.03 (1H, d), 7.68 (1H, d), 6.84 (1H, dd), 6.75 (1H, d), 6.20 (1H, d), 3.87–3.85 (2H, d), 1.72 (1H, m), 1.49 (2H, m), 1.39 (2H, m), 1.26 (2H, m), 1.19 (2H, m), 0.90 (3H, m), 0.85 (3H, m); NMR ^13^C (300 MHz, in CDCl_3_) δ 177.98, 164.24, 158.49, 155.60, 126.79, 118.08, 115.32, 112.34, 100.78, 71.21, 39.15, 30.36, 28.95, 23.72, 22.88, 13.83, 10.88.

#### 3.2.3. Biological Activities of Substituted Chromones

In this study, we have three chromones derivatives repellents represented by racemic 7-*sec*-butoxychromone (**10**), *R*-(−)-7-*sec*-butoxychromone (**14**), and *S*-(+)-7-*sec*-butoxychromone (**15**). Conversely, there are five attractants: racemic 7-*sec*-pentoxychromone (**11**), *R*-(−)-7-*sec*-pentoxychromone (**16**), *S*-(+)-7-*sec*-pentoxychromone (**17**), 7-*sec*-nonyloxychromone (**12**), and 7-(2′-ethyl)hexyloxychromone (**13**).

##### Chromone Derivatives with Repellent Activities against *Aedes albopictus*

Mosquito activity during testing was high AI > 70%. For racemic 7-*sec*-butoxychromone (**10**), *R*-(−)-7-*sec*-butoxychromone (**14**), and *S*-(+)-7-*sec*-butoxychromone (**15**), there were significant differences between the average number of mosquitoes in the control and treated zones. Averages recorded in the control were significantly higher than those of the treated zone (*p*-values < 0.05) except for the two quantities 5 and 10 mg of *S*-(+)-7-*sec*-butoxychromone (**15**) where the reverse was observed, i.e., the mean of mosquitoes in the control was significantly lower compared to that of treated (Table 16).

Racemic (**10**) and compound *R*-(−)-7-*sec*-butoxychromone (**14**) are repellent at any quantity. Compound *S*-(+)-7-*sec*-butoxychromone (**15**) is attractive at low quantity and repellent at high quantity. It is noted that the racemic has an effect corresponding substantially to the average of the effects of the two enantiomers. Maximum reported repellent effect is around 80% at 30 mg quantity of *R* enantiomer compound (Figure 5).

Comparison of the repellent effect of *R*-(−)-7-*sec*-butoxychromone (**14**) with those of known repellents at equivalent quantities 5, 10, and 30 mg shows that this new repellent is more effective than picaridin and DEET, which are considered to be the most effective products on the market (Table 17).

##### Chromone Derivatives with Attractant Activities towards *Aedes albopictus*

The mean activity of mosquitoes during testing of the three compounds, racemic 7-*sec*-pentoxychromone (**11**), *R*-(−)-7-*sec*-pentoxychromone (**16**), and *S*-(+)-7-*sec*-pentoxychromone (**17**), was >85%. Average numbers of mosquitoes recorded in the control zone were significantly lower than those of the treated zone (*p*-values < 0.05) except for the 30 mg of the two enantiomers *R*(**16**) and *S*-(+)-7-*sec*-pentoxychromone (**17**) (*p*-values > 0.05) where the mean of mosquitoes in the control was statistically similar to that of treated (Table 18).

Racemic 7-*sec*-pentoxychromone (**11**) and its enantiomers have attractive effects on *Aedes albopictus*. This attractive activity was observed for all quantities but there was an optimal quantity to have a maximum attractant effect. This corresponds to the quantity of 10 mg for all three compounds. Beyond this quantity, there was a clear decrease of the attractant effect as recorded for 30 mg. In general, racemic (**11**) and *R*-(−)-7-*sec*-pentoxychromone (**16**) had a better attractive activity than *S*-(+)-7-*sec*-pentoxychromone (**17**). Maximum attractant effect was around 63% at the quantity 10 mg of *R* enantiomer compound (Figure 6).

7-*sec*-nonyloxychromone (**12**) and 7-(2′ethyl)hexyloxychromone (**13**) are also among the chromone derivatives that have attractive activities on *Aedes albopictus*. The average numbers of mosquitoes recorded in the control zone were significantly lower than those of the treated zone (*p*-values < 0.05) at any quantity of these compounds (Table 19). We had an activity index > 70%.

The higher attractive effect for 7-*sec*-nonyloxychromone (**12**) was observed at 5 mg with KI = 57%. Beyond this quantity, we had a significant decrease in effect that goes down to KI = 18% at 30 mg. For 7-(2′ethyl)hexyloxychromone (**13**), maximum attractive effect was observed at a low quantity (0.5 mg) with KI = 50%. This effect gradually decreased with increasing quantity. Beyond the quantity 10 mg, there was a plateau with KI around 25% (Figure 7). The compound 7-*sec*-nonyloxychromone (**12**) had a slightly better attractive activity than 7-(2′ethyl)hexyloxychromone (**13**).

The attractive activities of the two products, 7-*sec*-pentoxychromone (**11**) and 7-*sec*-nonyloxychromone (**12**), were compared with that of octenol (known attractant) at equivalent quantities. At a low quantity (5 mg), 7-*sec*-nonyloxychromone (**12**) and 7-*sec*-pentoxychromone (**11**) had a better attractant effect than octenol with KI > 57%. On the other hand, at a high quantity (10 and 30 mg), octenol had the highest attractant effect with KI > 54%. Generally, 7-*sec*-pentoxychromone (**11**) had a high attractant effect compared to 7-*sec*-nonyloxychromone (**12**) (Table 20).

The combination of the new attractant, 7-*sec*-nonyloxychromone (**12**), with 4-hydroxycoumarin (recently found to be a selective attractant compound of *Aedes albopictus*) [15] was carried out to evaluate a possible synergistic effect between them. At a low quantity, 4-hydroxycoumarin had an attractant effect with a KI of the order of 40%. A 10% increase in attractiveness was recorded for higher quantities (10 and 30 mg) of this compound. A similar effect was observed for 7-*sec*-nonyloxychromone (**12**) with a Kairomone index KI = 57% but a significant decrease in this effect was observed from the quantity 10 mg. The use of a 50/50 mixture of 4-hydroxycoumarin and 7-*sec*-nonyloxychromone (**12**) had a greater attractant power than the two compounds used separately for all quantities tested (Table 21). This effect was more stable with a kairomone index between 55% and 62% (Figure 8). Indeed, the 7-*sec*-nonyloxychromone (**12**) was more efficient at a low quantity although the 4-hydroxycoumarin was more efficient at a higher quantity.

## 4. Conclusions

The results described in this article were obtained in our joint laboratory in Madagascar and are linked to sustainable development.

In the field of industrial ecology, we synthesized CNSL oxyacetate (**8**) without any separation or purification in a single step and with a good yield. This product has surfactant property similar to those of LABSA and SLES but at a lower concentration. It also has foaming and detergent properties similar to commercial fossil-based surfactants (LABSA and SDS). In addition, sodium oxyacetate of CNSL (**8**) exhibits significantly lower toxicity and ecotoxicity. Biodegradation has yet to be investigated, but CNSL itself is known to exhibit rapid biodegradation [37]. By using a polluting agro-industrial waste and a relatively simple process, we obtained a surfactant with similar or better functional properties and a lower ecological impact compared to the most widely used industrial surfactants. Considering the availability of this natural resource, this appears as an ecological alternative to several basic chemicals and a pilot plant is underway.

The use of chemical ecology could be an alternative to insecticide in many ways. Our results show for the first time that chromone derivatives have significant attractive or repellent properties on *Aedes albopictus* depending on the structure of the substituent (chain length, stereochemistry, etc.). The repellent effect of the *R*-enantiomer is as strong as DEET and picaridin in identical amounts. Secondary alkoxy groups higher than butyl (pentyl and nonyl) exhibit attractive properties. In comparison with 3-octenol [38], some alkoxychromones have a greater attractiveness in low quantities. Based on the results, an additive or synergetic effect was observed between 4-hydroxycoumarin and 7-*sec*-nonyloxychromone.

The products derived from 7-hydroxychromone (**9**) have high molecular weight, therefore a low vapor pressure allows them to have a lasting effectiveness in terms of repellency and attractiveness.

Repellent molecules can protect the exposed population and attractant molecules can be used for the construction of selective traps for dangerous species. In natura evaluation of our products are in preparation in the Madagascar rainforest.

We believe that we (and others) have shown that useful research can be carried out in developing countries, even if the objectives, technologies and strategy must be adapted to the social, technical, and economic conditions of these countries. The principles and methods derived from “green chemistry” are particularly useful in this field.

## Data Availability

The data presented in this study are available on request from the corresponding author. The data are not publicly available due to patent in process.

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
