# Peer review of "Development of Sustainable Chemistry in Madagascar: Example of the Valuation of CNSL and the Use of Chromones as an Attractant for Mosquitoes"

_molecules, 2021, doi:10.3390/molecules26247625_

Round 1
Reviewer 1 Report
Dear Author,
I have gone through the whole manuscript. No doubt it's nice piece of work. I have attached few of my comments on the attached file. Moreover, I have the following queries or suggestions to improve this manuscript to publish in Molecules.
[1] Title should be improved by deleting atleast one ''Chemistry'' term
[2] Similarity index of the manuscript should added.
[3] Should be consistent in using any term throughout the manuscript
[4] I don't understand what are the difference between section 2.1 & Section 3. Why the manuscript has two materials & methods section?
[5] Conclusion is too long, please write it based on your major findings, draw back & future recommendation.
[6] Please strictly follow the styles of the journal Molecules in preparing the manuscript.
[7] The citation numbers in the text and the number used for identifying compound is sometimes same. It make confusion and may mislead the readers. Please improve it.
[8] Lots of typos & Grammartical errors throughout the manuscripts, please correct those.

Reviewer 2 Report
The article “Sustainable Chemistry: An Opportunity for the Development of Chemistry in Madagascar". It is recommended that this paper may be resubmitted to the journal after a major revision:
- The introduction needs to improve. The motivation of work is unattractive. The Introduction section is not well cited.
- In the experimental part Separation of CNSL constituents needs more details
- The abstract is not comprehensively written and also lacks the originality value of the paper. The abstract needs to be added purpose of the research, material, and methods adopted, and practical implications of the research. The abstract should be revised appropriately which makes clear.
- The Materials and Methods part needs more explanation such as the use and purity of chemicals.
- Which are the uncertainties associated with the data shown in Tables 7-9 and Figure 2?
- The conclusion should be concisely present the results of the paper.
- The solubility of CNSL oxyacetic acid is higher than pure acardic acid oxyacetic acid in both water and hexane. Needs some clarifications.
- Aquatic toxicity was tested with Nauplii (Artemia catvis cysts). What are the advantages of this method?
- The research approach adopted is not so clear which leads to implausibility. How did you compare your results with the previous researches? How your results are better than the previous ones. It would be very convenient to compare the results with well-known systems.
- More explanation/clarification/significance is needed for Figs 2, and 3
- There are some typos in the manuscript. Please correct them in the revised manuscript.
- What is the social contribution of your research?
- Concluding remarks lack limitations and future directions of the research. Indicate the limitation of your research and suggest a roadmap for future research.
Round 2
Reviewer 1 Report
Dear Authors,
Thank you so much for all the corrections. But still the title seems incomplete, & something is missing.
Thanks.
Author Response
Thank you.
Reviewer 2 Report
The manuscript was improved and now could be accepted
Author Response
Thank you.